# Pulmonary Complications after Vertebral Body Tethering: Incidence, Treatment, Outcomes and Risk Factor Analysis

**DOI:** 10.3390/jcm11133778

**Published:** 2022-06-29

**Authors:** Per Trobisch, Filippo Migliorini, Thomas Vanspauwen, Alice Baroncini

**Affiliations:** 1Department of Spine Surgery, Eifelklinik St. Brigida, Kammerbruchstr. 8, 52152 Simmerath, Germany; per.trobisch@artemed.de (P.T.); thomas.vanspauwen@artemed.de (T.V.); 2Department of Orthopaedics and Trauma Surgery, RWTH Aachen University Clinic, 52074 Aachen, Germany; migliorini.md@gmail.com

**Keywords:** vertebral body tethering, pleural effusion, pulmonary complications, complication management, adolescent idiopathic scoliosis, risk factor analysis

## Abstract

Introduction: Vertebral body tethering (VBT) is gaining popularity for the management of selected AIS patients. The most frequent non-mechanical complications after VBT are pulmonary complications, with a reported incidence of up to 8% for recurrent pleural effusion. However, only trace data have been published on this topic. We aimed to analyze the incidence, timing, treatment, outcomes and risk factors of pulmonary complications after VBT. Materials and Methods: All patients who underwent VBT between September 2018 and September 2022 were retrospectively reviewed. The rate of pulmonary complications was analyzed and the symptoms, timing of onset, treatment and outcomes were recorded. An analysis of demographic, radiographic, surgical and pulmonary function data was conducted to explore possible risk factors for pulmonary complications. Results: Data from 140 patients were available: 14 experienced a pulmonary complication 1 day to 6 weeks after VBT, with 9 presenting a recurrent pleural effusion. A total of 13 patients required invasive treatment. All recovered without sequelae. The risk factor analysis did not result in any significant observations. However, 11/14 patients had had a diaphragm split. Conclusion: Pulmonary complications were observed in 10% of patients. The timing, symptoms and required treatment were heterogeneous. Pleural effusion seems to be more common after diaphragm crossing, but evidence is not yet conclusive.

## 1. Introduction

Vertebral Body Tethering (VBT) is increasingly and rapidly becoming popular as a non-fusion surgical alternative for selected patients with adolescent idiopathic scoliosis (AIS). While VBT was originally developed as a growth-modulating technique for skeletally immature patients [1,2], recent evidence shows its efficacy as a correction technique also in patients approaching skeletal maturity [3,4]. Several studies have observed success rates above 80% when success is defined as a patient with a controlled scoliosis, ideally below 30°, two or more years after VBT or at skeletal maturity [5,6,7,8]. However, the complication rate after VBT is still high. Many complications are mechanical and possibly related to patient selection [3,5,6,7,8,9,10,11,12]. Non-mechanical perioperative complications, though not uncommon [5,6,9], are currently less enthusiastically discussed. Non-mechanical perioperative complications include rarities such as ureteral injury or screw misplacement resulting in penetration of the spinal canal (liquor deficiency syndrome) [6,13,14,15], but the most frequent non-mechanical complications are intrathoracic and often represented by pleural effusion [6]. Although several authors have observed pleural effusions after VBT, no study has yet focused on this complication. With an incidence of up to 8% [16], it is not unlikely that surgeons who are planning to start VBT will observe pleural effusions or other pulmonary complications. We therefore sought to analyze data on pulmonary complications among our patients and to report details about the incidence, diagnostic tools, treatment options, risk factors and outcomes that we experienced.

## 2. Methods

This single-center, single-surgeon retrospective study was conducted according to the STROBE statement [17].

### 2.1. Patient Recruitment

All consecutive patients who underwent VBT at our institution from September 2018 to September 2021 were considered for inclusion. Only patients who presented a complete pre- and perioperative diagnostic including preoperative and 1st standing X-ray and preoperative pulmonary function were included in this study.

### 2.2. Surgical Technique and Postoperative Care

Thoracic curves were usually performed in a combination of one or two mini-open intercostal approaches (5 to 6 cm each) and one or two thoracoscopic portals for video-assisted thoracic surgery (VATS). Thoracolumbar/lumbar curves were instrumented by using two mini-open approaches: a retroperitoneal one to L2 and below and an intercostal one to L1 and above. The diaphragm was usually not detached but punctured to let the tether pass through (diaphragm split). Double curves were all operated via a single-staged approach with the thoracolumbar/lumbar curve being instrumented first.

All patients received a postoperative chest tube, one for each side for double instrumentation. The tube was removed once the output decreased to less than 100 mL per 24 h. Once this threshold was reached, as an extra safety measure, we performed ultrasounds to quantify the remaining pleural effusion (Figure 1): the chest-tube was removed if less than 200 mL residual effusion was measured. The patients were monitored at the Intermediate Care Unit (ICU) until the chest tubes were removed.

From the first postoperative day, patients were encouraged to train with a tri-flow multiple times per day and were asked to continue with the training for 4–6 weeks after hospital discharge. If patients presented symptoms such as dyspnea or cough, a chest X-ray was performed along with a pleural sonography, which was repeated daily until symptoms receded or to control the efficacy of the adopted therapy. Patients with acute symptoms were taken back to the ICU or to surgery. Patients with respiratory symptoms were discharged after symptoms subsided and when the remaining pleural effusion was less than 100 mL in the sonograph.

While there is no established protocol for the treatment of pleural effusion, at our institution we tend to treat these patients conservatively (ultrasound controls) when symptoms are absent or mild, such as a light cough or minor dyspnea, when the fluid collection is less than 200 mL as measured in pleural ultrasounds and when the collection is regredient within 24–48 h. Otherwise, invasive treatment is indicated. This usually consist in the reinsertion of the chest tube and dietary restriction in case of chylothorax. When an active bleeding is suspected, a surgical revision is usually performed.

Routine follow-up visits are conducted at 6 weeks, 6 months and at 1, 2 and 5 years postoperatively. At each timepoint, along with a clinical assessment, whole spine standing X-rays and pulmonary function are taken. At the 1-, 2- and 5-year follow-ups, the SRS-22 questionnaire is given to the patient as well. The level of physical activity is controlled at the 1-year follow-up with the sport activity questionnaire [18].

### 2.3. Outcomes of Interest

Although pulmonary complications usually occur within the first three months after surgery, patients were monitored for this complication throughout the follow-up (e.g., pulmonary function tests, specific questions at follow-up visits). The rate, type and timing of pulmonary complications were recorded, along with the correlated symptoms and therapy.

Demographic, radiographic (curve type, coronal and sagittal parameters from the preoperative and 1st standing X-ray) and intraoperative (anesthesia time and surgical time, instrumented levels) data were collected, along with the preoperative pulmonary function data, to seek possible risk factors for the development of a pulmonary complication. The curve type was defined following a previously published algorithm [19]. Regarding the instrumented levels, the lowest instrumented vertebra (LIV) for thoracic curves and the upper instrumented vertebra (UIV) for lumbar curves were noted. Regarding the pulmonary function parameters, a clinically significant difference was defined as a change of ±10% [20].

### 2.4. Statistical Analysis

The statistical analyses were performed using the software IBM SPSS 25. Regarding the analysis of the risk factors, the mean difference (MD) effect measure was evaluated with the *t*-test to assess statistical significance for continuous data. For binary data, the odds ratio (OR) effect measure was performed with the χ^2^ test to assess statistical significance. The confidence interval (CI) was set at 95% in all comparisons. Values of *p* < 0.05 were considered statistically significant.

## 3. Results

### 3.1. Patient Recruitment and Baseline Data

During the observation period, VBT was performed on 140 patients. The pre- and perioperative records were complete and available for all subjects, so that 140 patients were included in the study. The mean follow-up was 18.6 months.

Twenty-two patients were male (15%) and the mean age was 15.7 ± 3.9 years old. The mean Risser grade was 2.9 ± 3.9, and the mean Sanders score was 5.9 ± 1.8. Preoperatively, the thoracic curves measured averagely 54.4 ± 17.6° and bent down to 34.2 ± 17.5° on side-bending X-rays. The lumbar curves measured 48.3 ± 14.3° and bent to 19.7 ± 15.6°. The mean thoracic kyphosis was 32.9 ± 13.4° and the mean lumbar lordosis was 52.7 ± 11.6°. At the first standing X-ray after surgery, the mean thoracic Cobb angle measured 25.2 ± 10° (mean correction 50.7%) and the lumbar Cobb angle measured 16.7 ± 10° (mean correction 64%).

Regarding the pulmonary function, the mean Total Lung Capacity (TLC) was 102 ± 18%, the mean Forced Expiratory Pressure in 1 Second (FEV1) was 88.9 ± 16.3% and the Forced Vital Capacity (FVC) was 102.1 ± 20.5%.

### 3.2. Pulmonary Complications

Pulmonary complications were observed in 14 out of 140 included patients (10%). The most observed complication was a recurrent pleural effusion (nine cases, 64%), followed by haematothorax (two cases, 15%), pleural empyema (one case, 7%), chylothorax (one case, 7%) and contralateral atelectasis after right thoracic surgery (one case, 7%). The pulmonary complication was diagnosed one day to six weeks after index surgery.

For four patients, the pulmonary complication occurred while they were still inpatients after VBT. The remaining 10 patients were diagnosed after discharge: two were re-admitted at our hospital, and the other eight were treated in hospitals nearby their residence. The treatment of pleural effusions differed between these institutions, but all patients kept updating us and we therefore have a 100% follow-up rate.

All patients fully recovered without any remaining complaints. A preoperative and postoperative pulmonary function test was available for 7 of the 14 patients. The mean preoperative TLC was in line with the expected values for healthy patients of the same age (107.9 ± 17.2%), both the FEV1 and FVC were reduced (83 ± 9% and 80.7 ± 3.8%, respectively). One year after surgery, however, there was no clinically significant reduction in the pulmonary function parameters in comparison to the preoperative values (TLC 98.7 ± 7.1%, FEV1 104.5 ± 8.8% and FVC 81.5 ± 4.5%).

A detailed description of the patients who presented a pulmonary complication is shown in Table 1.

Examples of the X-rays of the two asymptomatic patients are shown in Figure 2 and Figure 3. Figure 4 shows the evolution of patient n. 6, who presented a chylothorax.

### 3.3. Risk Factor Analysis

The overview of all considered risk factors is shown in Table 2. The only parameters that showed a significant association with the risk of developing a pulmonary complication were the lumbar lordosis and the forced vital capacity.

The number of patients who developed a pulmonary complication according to the curve type or to the level of the thoracic UIV/lumbar LIV are reported in Table 3 and Table 4.

## 4. Discussion

This is the first study to offer a detailed analysis of pulmonary complications after VBT, a not-so-rare occurrence after this kind of surgery. We reported a 10% incidence of pulmonary complications and observed that the majority were diagnosed after discharge from the hospital, usually between 2 and 6 weeks postoperatively. As the number of VBT cases is increasing worldwide, we believe this study presents important data from which surgeons could benefit.

Just two of the considered risk factors, the lumbar lordosis and the FVC, showed a significant association with the development of pulmonary complications. However, it is unlikely that these parameters have any clinical significance and are thus interpreted as mathematical associations only. While the Risser grade was higher among patients who presented a pulmonary complication, this finding is not coherent with data on age and Sanders stage. Thus, we believe that this is likely to be a spurious correlation.

Despite the high rate of pulmonary complications, it is important to highlight that the affected patients recovered well and did not suffer any long-term complications. As many of our patients come from abroad, some of them did not have the opportunity to obtain pulmonary function tests and some have not yet reached a 1-year follow-up. For these reasons, the number of longer-term follow-ups is limited. However, the available data are in line with the postoperative pulmonary function values observed one year after VBT (TLC 99%, FEV1 89% and FVC 86%) [21].

The observation interval excludes the learning curve, as the surgeon performing all procedures had already performed over 50 VBT cases before September 2018. Cases prior to this date were not considered for the present study, as we had not yet routinely obtained a preoperative pulmonary function test for our patients. Thus, it is safe to say that the learning curve did not have any effect on the incidence of pulmonary complications in this cohort. Over time, the surgical technique was modified in an effort to reduce the rate of pulmonary complications. These modifications included a very limited opening of the pleura (only for the screw entry point), suturing of the diaphragm when a split is performed, and changes in postoperative chest tube management. In fact, at the beginning of our experience with VBT, the tubes were removed when the output was below 200 mL per 24/h. However, data are not sufficient to say whether these modifications had any influence on the rate of pulmonary complications.

The exact pathophysiology of this complication still raises questions. While hemo- or chylothorax seem to be directly related to the surgical procedure, a delayed pleural effusion may be related to postoperative activity. In our practice, we do not restrict postoperative activity and it has been observed that 94% of patients were able to resume their pre-operative activity level within three months after VBT [22]. However, most pleural effusions were diagnosed within the first month after surgery, when patients had probably not yet returned to their normal level of activity. Thus, limited mobility may play a role and patients should be advised to continue pulmonary training with the tri-flow even after hospital discharge.

Interestingly, only four cases of pulmonary complications occurred after a thoracic-only VBT (one in type 3 and 4 curves, respectively, and two in type 5 curves). Bypassing the diaphragm with a tether (double and thoracolumbar curves) may therefore represent a risk factor, as 10 cases out of 14 (71%) presented a pulmonary complication after a lumbar or bilateral procedure. However, we do not have an explanation as to why the majority of pleural effusions in double curves were diagnosed on the right side, where the diaphragm usually remained intact. Furthermore, the number of observations is still too small to reach a definitive conclusion on this point.

Similar considerations can be made for the analysis of the UIV/LIV. The majority of the patients presenting a pulmonary complication had the UIV/LIV at T11, but numbers are too small to reach a definitive conclusion and we do not have an explanation for this finding.

Alanay et al. observed 4 pulmonary complications in 31 analyzed patients, i.e., an incidence of 12% [9]. All patients in their series had a thoracic curve but six patients received VBT for a long thoracic curve, which usually requires splitting or dissection of the diaphragm. None of the patients had a bilateral VBT. One patient in their series was found to have a chylothorax, which required non-invasive dietary precautions, and one patient with a delayed effusion required readmission to the hospital and non-invasive medical treatment. In other similar-sized cohorts, Hoernschmeyer et al. reported 1 case of pneumothorax among 29 treated patients (3%) [10], while Samdani et al. observed 1 case of persistent atelectasis in 32 patients (3%) [23]. In smaller-sized groups, Newton observed 1 atelectasis case in 23 operated patients (4%) [24] and Pehlivanoglu reported 1 case of chylothorax in 21 treated subjects (5%) [25]. Wong et al. observed 5 pulmonary complications in 3/5 patients (60%) [26]. Conversely, Boudissa and colleagues did not observe any pulmonary complications in six operated patients. While many of the studies available in the literature showed a lower rate of pulmonary complications than the one presented in this study, other authors mainly performed thoracic VBT and the patient cohorts are thus not directly comparable.

The Humanitarian Use Device Exemption study, which was published by the US Food and Drug Administration, and which resulted in approval of the first VBT device in the US, reported a 5.3% incidence (3/57 patients) in a very homogeneous patient cohort consisting of only thoracic Lenke type 1 curves. No information was provided with respect to the treatment or outcomes [27]. Comparing these data to the ones of the presented cohort supports the hypothesis that diaphragm splitting may represent a risk factor for pulmonary complications. Further studies on a larger patient cohort will be required to clarify this point.

Rushton et al. reported 2 out of 112 patients with a pleural effusion after VBT [15]. While their calculated incidence was significantly lower compared to our patient population, most patients in this study received a single curve VBT. Only four patients from their cohort had a double curve VBT and all these four patients were operated on in a staged manner. Therefore, no patient had a single-staged double curve VBT. However, the number of observations for double instrumentation presented in Rushton’s study is too small to support the hypothesis that single-staged surgery represents a risk factor for the development of pulmonary complications. Both patients in Rushton’s series required invasive treatment; one had a chest drainage and the other patient required surgical revision for bleeding control [15].

With 140 analyzed patients, to our best knowledge, this study has the highest number of included patients who have had VBT. Additionally, the study reviews a single surgeon’s patients and therefore the surgical technique confounder can be limited to a minimum. On the other hand, this study does not come without limitations, which are not only related to the retrospective methodology. While we are able to calculate the incidence of pleural effusions after VBT and hypothesize a few risk factors, we still have no insight into the exact pathophysiology and therefore have not been able to eradicate this complication in our practice. Nevertheless, with this analysis of a very heterogeneous patient cohort we are able to add valuable information about a not-so-rare complication after VBT.

## 5. Conclusions

In the presented cohort, an incidence of pulmonary complications after VBT of 10% was observed. The timing, symptoms and required treatment were heterogeneous. However, none of the presented patients showed long-term consequences. Pleural effusion seems to be a common complication after diaphragm crossing VBT, but evidence is not yet conclusive. Patients need to be informed that this complication can occur after discharge from the hospital and that it may require re-admission. However, patients can be reassured that long-term consequences are unlikely.

## Figures and Tables

**Figure 1 jcm-11-03778-f001:**
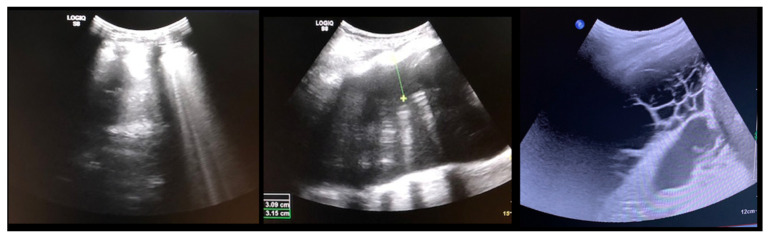
(**Left**) normal pleural ultrasound with the lung-shadow attached to the ribs; (**Middle**) Pleural effusion shown by a black/empty area between the ribs and the lung tissue; (**Right**) Cavernous hematoma shown by multiple cystic formations.

**Figure 2 jcm-11-03778-f002:**
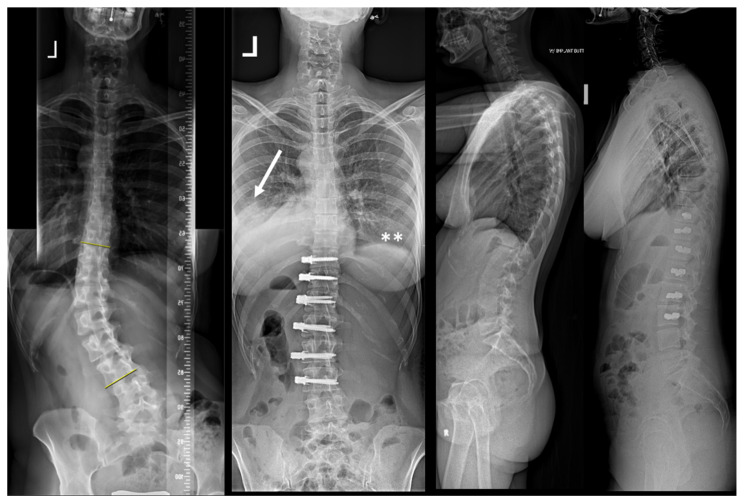
The patient (n. 8 in the series) presented with a lumbar curve that measured 41° but resulted in a severe coronal imbalance and the patient suffered from daily pain. Surgery was able to almost completely correct her deformity. Pleural effusion on the left was diagnosed on routine post-operative radiographs with a vanished lateral recess (white arrow; ** shows a visible right lateral recess for comparison).

**Figure 3 jcm-11-03778-f003:**
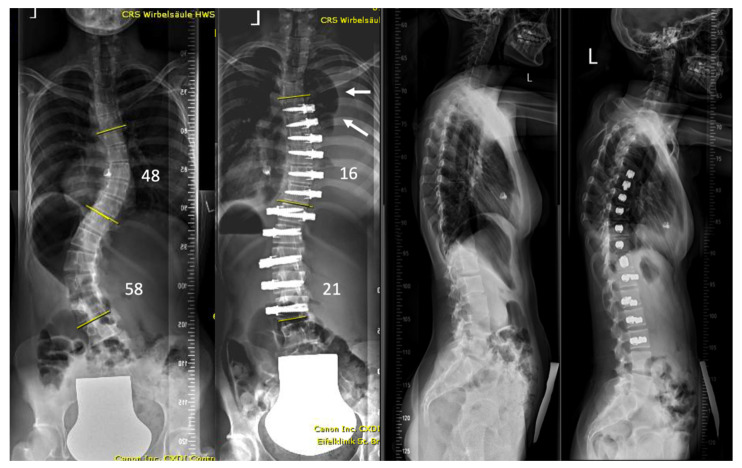
17-year-old but very small patient (34 kg) with mild form of Di-George syndrome, which is known to present with vessel anomalies. Post-operative recovery was uneventful with no symptoms of fatigue or shortness of breath. Severe pleural effusion was noticed on first erect postoperative spine radiograph (arrows).

**Figure 4 jcm-11-03778-f004:**
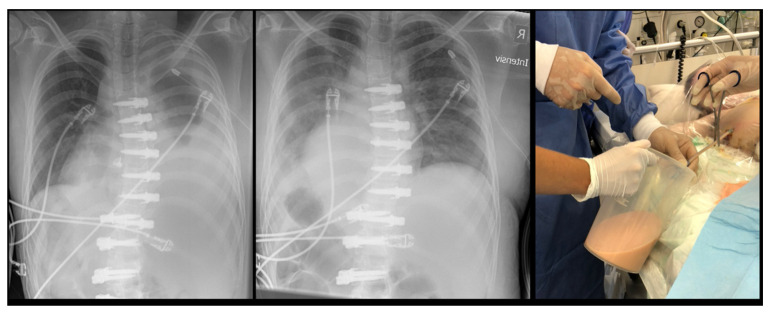
Patient from Figure 3. (**Left**) Chest radiograph after drainage of 1 L; (**Middle**) Chest radiograph after drainage of 2.5 L; (**Right**) Macroscopic appearance of orange-milky chylos.

**Table 1 jcm-11-03778-t001:** Overview of the characteristics of the patients who presented pulmonary complications, with details regarding presentation symptoms, time to diagnosis and treatment. CRP = C reactive protein.

Patient	Age (Years)	Sex	Curve Type	Instrumented Levels	Complication	Time to Diagnosis	Side of the Complication	Symptoms	Treatment
1	13.3	F	2	T5-T12 right T12-L4 left	Pleural effusion	2 weeks	Right	Minor dyspnea	Ultrasound, conservative treatment
2	17.6	F	4	T6-L1 right	Pleural effusion	2 weeks	Right	Dyspnea, fatigue	2 × aspiration
3	16.2	F	4	T5-T12 right	Contralateral atelectasis	2 days	Left	Severe dyspnea	Re-intubation for 3 days, 3 bronchoscopies and removal of a mucus plug
4	17.6	M	1	T9-L3 right	Pleural effusion	4 weeks	Right	Chest pressure	Chest tube reinsertion
5	16.2	F	2	T5-T11 right T11-L3 left	Pleural effusion	3 weeks	Bilateral	Chest pain and elevated CRP levels	Bilateral aspiration, forced diuresis and i.v. albumin treatment
6	17.7	F	2	T5-T11 right T11-L3 left	Chylothorax	3 days	Right	None effusion, diagnosed on routine post-op X-ray	Chest tube reinsertion and dietary restriction
7	16.8	M	2	T5-T11 right T11-L4 left	Pleural effusion	3 weeks	Left	Unknown	Explorative thoracoscopy
8	17.9	F	1	T10-L3 left	Pleural effusion	4 days	Left	None, effusion diagnosed on routine post-op X-ray	Aspiration followed by chest tube reinsertion for recurrent effusion
9	14.6	F	2	T5-T11 right T11-L4 left	Pleural effusion	3 weeks	Right	Fatigue, dyspnea	Chest tube reinsertion, antibiotics for co-existing pyelonephritis
10	14.3	F	1	T11-L4 left	Pleural effusion with concomitant infection	3 weeks	Left	Sudden sharp pain in the left chest and dyspnea	Attempted aspiration and chest tube without output. VATS and six weeks antibiotitcs because of postivie culture for staph epidermidis
11	12	F	4	T5-T11 right	Haematothorax	1 day	Right	No symptoms, significant blood loss noticed after declamping the chest tube and drop of haemoglobin levels	Emergency explorative thoracotomy using the same surgical approach. No active bleeding found but clotted hematoma
12	13	F	2	T6-T12 right T12-L4 left	Haematothorax	6 weeks	Right	Acute chest pain	Emergency explorative thoracotomy
13	16.5	M	1	T10-L4 left	Pleural empyema	5 weeks	Left	Dyspnea, elevated CRP levels	VATS and antibiotic therapy
14	16.3	F	2	T5-T11 right T11-L4 left	Pleural effusion	5 weeks	Right	Dyspnea	Aspiration

**Table 2 jcm-11-03778-t002:** Summary of the risk factor analysis for developing a pulmonary complication after VBT. MD = mean difference; OR = odds ration; SE = standard error, CI = confidence interval.

Endpoint	No Pulmonary Complications (N = 126)	Pulmonary Complications (N = 14)	MD/OR	SE	95% CI	*p*
*Demographic data*						
Age	15.8 ± 4.8	15.7 ± 1.8	2.6	1.4	−0.2 to 5.4	0.1
Gender (male)	15% (19 of 126)	21.4% (3 of 14)	0.6	0.9	0.1 to 2.3	0.5
Risser	2.8 ± 1.9	3.3 ± 1.9	−1.5	0.6	−2.6 to −0.4	0.007
Sanders	5.9 ± 1.9	6.2 ± 2.0	0.3	0.6	−0.7 to 1.3	0.6
*Preoperative data*						
Thoracic Cobb angle (°)	53.8 ± 17.7	47.9 ± 16.1	−5.9	5.0	−15.7 to 3.9	0.2
Thoracic bending (°)	35.0 ± 17.6	26.0 ± 13.7	−9.0	5.0	−18.9 to 0.9	0.08
Thoracic flexibility (%)	39.0 ± 20.8	48.2 ± 16.2	9.2	6.0	−2.5 to 20.9	0.1
Lumbar Cobb angle (°)	48.3 ± 14.1	48.0 ± 15.3	−0.3	4.1	−8.4 to 7.8	0.9
Lumbar bending (°)	19.9 ± 15.6	19.5 ± 14.6	−0.4	4.5	−9.3 to 8.5	0.9
Lumbar flexibility (%)	61.2 ± 37.1	64.1 ± 24.9	2.9	10.5	−17.9 to 23.7	0.8
Thoracic kyphosis (°)	33.0 ± 13.7	31.2 ± 9.8	−1.8	3.9	−9.5 to 5.9	0.6
Lumbar lordosis (°)	53.0 ± 11.5	48.8 ± 12.2	−4.2	3.4	−10.8 to 2.4	0.2
Sagittal vertical axis (mm)	4.8 ± 27.1	6.2 ± 23.2	1.4	7.8	−14 to 16.8	0.9
Coronal balance (mm)	9.2 ± 18.9	8.9 ± 24.0	−0.3	5.6	−11.4 to 10.8	0.9
Pelvic incidence (°)	50.3 ± 13.7	45.7 ± 15.3	−4.6	4.0	−12.5 to 3.3	0.3
Pelvic tilt (°)	9.4 ± 7.5	13.3 ± 15.5	3.9	2.5	−0.9 to 8.7	0.1
*Intraoperative data*						
Double tether	53% (67 of 126)	42.8% (6 of 14)	1.3	0.5	0.4 to 4.1	0.7
Disk release	66% (84 of 126)	57% (8 of 14)	1.2	0.3	0.3 to 3.9	0.7
Anaesthesia time (*min*)	334.9 ± 93.3	335.0 ± 87.2	0.1	2.7	−53.3 to 53.5	0.9
Surgical time (*min*)	236.0 ± 28.8	232.5 ± 71.3	−3.5	10.1	−23.4 to 16.4	0.7
*1st erect X-ray*						
Thoracic Cobb angle (°)	25.7 ± 10.2	21.1 ± 6.0	−4.6	2.9	−10.3 to 1.1	0.1
Thoracic correction (%)	50.7 ± 15.6	50.5 ± 17.9	−0.2	4.6	−9.3 to 8.9	0.9
Lumbar Cobb angle (°)	16.6 ± 9.9	17.7 ± 11.2	1.1	2.9	−4.6 to 6.8	0.7
Lumbar correction (%)	64.5 ± 20.7	63.9 ± 16.1	−0.6	5.9	−12.3 to 11.1	0.9
Thoracic kyphosis (°)	34.0 ± 11.7	36.2 ± 7.9	2.2	3.3	−4.3 to 8.7	0.5
Lumbar lordosis (°)	46.5 ± 10.4	39.9 ± 12.2	−6.6	3.1	−12.6 to −0.5	0.03
Sagittal vertical axis (mm)	26.2 ± 27.3	35.1 ± 27.0	8.9	3.1	−6.8 to 24.6	0.3
Coronal balance (mm)	15.5 ± 22.2	19.7 ± 20.9	4.2	6.4	−8.5 to 16.9	0.5
*Pulmonary function*						
Total lung capacity	101.3 ± 17.6	109.1 ± 19.8	7.8	5.2	−2.4 to 18	0.1
Forced expiratory volume 1 s	89.0 ± 16.8	87.3 ± 10.2	−1.7	4.8	−11.1 to 7.7	0.7
Forced vital capacity	101.0 ± 19.7	113.2 ± 23.8	12.2	5.9	0.6 to 23.7	0.04

**Table 3 jcm-11-03778-t003:** Summary of the events (pulmonary complication) according to the curve type (1 = thoracolumbar/lumbar curves, 2 = double curves, 3 = long thoracic curves, 4 = short thoracic curves, 5 = presence of a rigid, high thoracic curve).

Curve Type	Patients (N)	Pulmonary Complication
1	20.7% (29 of 140)	10.3% (3 of 29)
2	50% (70 of 140)	10% (7 of 70)
3	10.7% (15 of 140)	6.6% (1 of 15)
4	15% (21 of 140)	4.7% (1 of 21)
5	3.4% (5 of 140)	40% (2 of 5)

**Table 4 jcm-11-03778-t004:** Summary of the events (pulmonary complication) according to the lumbar upper instrumented vertebra (UIV) and/or thoracic lowest instrumented vertebra (LIV).

UIV/LIV	Patients (N)	Pulmonary Complication
T10	18.5% (26 of 140)	7.6% (2 of 26)
T11	34.3% (48 of 140)	14.5% (7 of 48)
T12	28.5% (40 of 140)	7.5% (3 of 40)
L1	11.3% (13 of 140)	7.7% (1 of 13)

## Data Availability

Data can be made available upon reasonable request.

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
