# Peer review of "Pulmonary Complications after Vertebral Body Tethering: Incidence, Treatment, Outcomes and Risk Factor Analysis"

_jcm, 2022, doi:10.3390/jcm11133778_

Round 1

Reviewer 1 Report

Introduction

line 33 - no reference

what is the indication for VBT?

Truly it's the method for immature patients with scoliosis, with open triradiate cartilage; contraindicated in matured ones (Parent, Stefan MD, Ph.D.; Shen, Jesse MD, MSc; Anterior Vertebral Body Growth-Modulation Tethering in Idiopathic Scoliosis: Surgical Technique. Journal of the American Academy of Orthopaedic Surgeons: September 1, 2020 - Volume 28 - Issue 17 - p 693-69. DOI: 10.5435/JAAOS-D-19-00849)

Methods

The group of patients is not well described; few data are in Results and Methods (same subsection Patient Recruitment). 

The mean age at the time of surgery is almost 16 y, and the Risser test is 3 - I consider it as a treatment "out of label".

line 80 - no reference to the table

what is the protocol of postoperative assessment in the outpatient department? 

what examinations are performed to diagnose and control pulmonary complications?

Risk factor analysis - do not contain any statistical analysis (table 3 and 4), so You are not able to do any conclusion

table 2 doesn't contain any information about the risk factor, only the demographic ones, and pre and postop data

Discussion

too little data from the literature.

6 of 19 references are the auto citation.

Author Response

Thank you for the time invested in reviewing our manuscript. Please find the detailed answers to your questions and comments below.

Introduction

line 33 - no reference

Authors’ response:

Thank you for pointing this out. Unfortunately, the line numbering we have does not seem to correspond to yours and I thus cannot retrieve which line you are referring to. If you were so kind as to provide the phrase that is missing the reference, we will be happy to add it.

what is the indication for VBT?

Truly it's the method for immature patients with scoliosis, with open triradiate cartilage; contraindicated in matured ones (Parent, Stefan MD, Ph.D.; Shen, Jesse MD, MSc; Anterior Vertebral Body Growth-Modulation Tethering in Idiopathic Scoliosis: Surgical Technique. Journal of the American Academy of Orthopaedic Surgeons: September 1, 2020 - Volume 28 - Issue 17 - p 693-69. DOI: 10.5435/JAAOS-D-19-00849)

Authors’ response:

Thank you for your question. We agree that VBT was initially recommended for very immature patients (open TRC). However, more recent data show that too much residual growth (open TRC) can result in overcorrection. Many authors’ recommendations have therefore shifted to a later timing (Alanay et al, Thoracoscopic Vertebral Body Tethering for Adolescent Idiopathic Scoliosis: Follow-up Curve Behavior According to Sanders Skeletal Maturity Staging. Spine (Phila Pa 1976). 2020 Nov 15;45(22):E1483-E1492. doi: 10.1097/BRS.0000000000003643). A recent study has proposed a dual modality for VBT, one relying on growth modulation and one being an anterior correction technique (Bernard et al. Dual modality of vertebral body tethering : anterior scoliosis correction versus growth modulation with mean follow-up of five years. Bone Jt Open. 2022 Feb;3(2):123-129. doi: 10.1302/2633-1462.32.BJO-2021-0120.R1). We agree with this indication for VBT and employ it for immature patients and for patients approaching skeletal maturity as well (Risser 4/Sanders7). We included this information in the introduction, the added paragraph now reads as follows:

“While VBT has been originally developed as a growth-modulating technique for skeletally immature patients1,2 , recent evidence show it’s efficacy as a correction technique also in patients approaching skeletal maturity 3,4.”

Methods

The group of patients is not well described; few data are in Results and Methods (same subsection Patient Recruitment).

Authors’ response:

Thank you for raising this point. We agree that little baseline data was given for the entire cohort. These informations have been added to the manuscript, the added paragraph reads as follows:

“Twenty-two patients were male (15%), the mean age was 15.7 ± 3.9 years old. The mean Risser grade was 2.9 ± 3.9, mean Sanders score was 5.9 ± 1.8. Preoperatively, thoracic curves measured averagely 54.4 ± 17.6° and bent down to 34.2 ± 17.5° on side-bending x-rays. Lumbar curves measured 48.3 ± 14.3° and bent to 19.7 ± 15.6°. The mean thoracic kyphosis was 32.9 ± 13.4° and the mean lumbar lordosis was 52.7 ± 11.6°. At the first standing x-ray after surgery, the mean thoracic Cobb angle measured 25.2 ± 10° (mean correction 50.7%) and the lumbar Cobb angle measured 16.7 ± 10° (mean correction 64%).

Regarding the pulmonary function, the mean Total Lung Capacity (TLC) was 102 ± 18%,the mean Forced Expiratory Pressure in 1 Second (FEV1) was 88.9  ± 16.3% and the Forced Vital Capacity (FVC) was 102.1 ± 20.5%.”.

The mean age at the time of surgery is almost 16 y, and the Risser test is 3 - I consider it as a treatment "out of label".

Authors’ response:

Thank you for raising this point. Next to growth-modulating VBT, we perform this technique in patients approaching skeletal maturity as well, as this method is supported by the literature and allows for more predictable results (Alanay et al, Thoracoscopic Vertebral Body Tethering for Adolescent Idiopathic Scoliosis: Follow-up Curve Behavior According to Sanders Skeletal Maturity Staging. Spine (Phila Pa 1976). 2020 Nov 15;45(22):E1483-E1492. doi: 10.1097/BRS.0000000000003643; Bernard et al. Dual modality of vertebral body tethering : anterior scoliosis correction versus growth modulation with mean follow-up of five years. Bone Jt Open. 2022 Feb;3(2):123-129. doi: 10.1302/2633-1462.32.BJO-2021-0120.R1).
While the FDA approval in the USA limits the in-label use of VBT to skeletally immature patients, such restrictions are not present in Europe and therefore we have no limit of age or skeletal maturity to the in-label use of VBT implants.

line 80 - no reference to the table

Authors’ response:

Thank you for raising this point. Unfortunately, we are not able to reconstruct what line you are referring to. However, we have checked that all four tables are referenced in the text.

what is the protocol of postoperative assessment in the outpatient department?

Authors’ response:

Thank you for raising this point. Routine follow-up visits are conducted at 6 weeks, 6 months and at 1, 2  and 5 years postoperatively. At each timepoint, along with a clinical assessment, whole spine, standing x-rays and pulmonary function are taken. At the 1, 2 and 5 years follow-up the SRS-22 questionnaire is given to the patient as well. The level of physical activity is controlled at the 1-year follow-up with the sport activity questionnaire. These details have been added to the manuscript.

what examinations are performed to diagnose and control pulmonary complications?

Authors’ response:

Thank you for raising this point. Further details have been added to better explain the postoperative care in case of pulmonary complications. The added text reads as follows:
“All patients received a postoperative chest tube, one for each side for double instrumentation. The tube was removed once the output decreased to less than 100ml per 24 hours. Once this threshold was reached, as an extra safety measure we performed ultrasounds to quantify the remaining pleural effusion (Figure 1): the chest-tube was removed if less than 200ml residual effusion was measured. The patients were monitored at the Intermidiate Care Unit (ICU) as long as the chest tubes were not removed.

From the first postoperative day, patients were encouraged to train with a tri-flow multiple times a day and were asked to continue with the training for 4-6 weeks after hospital discharge. If patients presented symptoms such as dyspnea or cough, a chest x-ray was performed along with a pleural sonography, which was repeated daily until symptoms receeded or to control the efficacy of the adopted therapy. Patients with acute symptoms were taken back to the ICU or to surgery. Patients with respiratory symptoms were discharges after symptoms subsided and when the remaining pleural effusion was less than 100ml in the sonography.”

Risk factor analysis - do not contain any statistical analysis (table 3 and 4), so You are not able to do any conclusion

table 2 doesn't contain any information about the risk factor, only the demographic ones, and pre and postop data

Authors’ response:

Thank you for your comment. A risk factor is a characteristic, condition or other variable that increases the patient’s risk of developing a disease – in this case a pulmonary complication. Thus, demographic data, radiographic data and perioperative data have been analyzed to investigate whether any of those was a risk factor or not.

Continuous data are presented in Table 2. This is not only a list of characteristics, but mean difference/odds ratio, standard error and confidence interval have been calculated to investigate whether any of these baseline factor did represent a risk factor or not. Thus, we respectfully disagree that no informations on risk factors has been produced.

For discrete data (curve type, UIV), the percentages of pulmonary complications has been recorded in different groups. We agree with the reviewer that available data is not sufficient to draw a definite conclusion, and we have clearly highlighted this in the discussion and conclusion. However, as there is a paucity of data regarding the risk factors for pulmonary complications after VBT and as data suggest that most cases of pulmonary complucations happen in patients who undergo diaphragm-crossing surgery, we believe that these preliminary considerations are still valuable to the reader.

Discussion

too little data from the literature.

6 of 19 references are the auto citation.

Authors’ response:

Thank you for your comment. We agree that there is little literature data presented to compare our results. Unfortunately, most of the available VBT literature does not focus on the non-mechanical complications of VBT and very little data is available. To the best of our knowledge, no other paper is available on the risk factors for pulmonary complications after VBT, so that it is not possible to compare our data with those of other groups. All data from papers reporting informations on pulmonary complications has been added. The text now reads as follows and the number of citations is now 28:

“In other similar-sized cohorts, Hoernschmeyer et al. reported one case of pneumothorax among 29 treated patients (3%) 10, while Samdani et al. observed one case of persistent atelectasis in 32 patients (3%)23. In smaller-sized groups, Newton observed one atelectasis case in 23 operated patients (4%)24 and Pehlivanoglu reported one case of chylothorax in 21 treated subjects (5%) 25. Wong et al observed five pulmonary complications in 3/5 patients (60%)26. Oppositely, Boudissa and colleagues did not observe any pulmonary complication in six operated patients. While many of the studies available in the literature showed a lower rate of pulmonary complications than the one presented in this study, other authors mainly performed thoracic VBT and the patient cohorts are thus not directly comparable.”

Reviewer 2 Report

Overall the paper is well-written. Considering VBT is quite a new method to treat Scoliosis, especially in some countries. More publications related to VBT will benefit the surgeons to avoid complications.

The authors explained in introduction "Non-mechanical peri-operative complications, despite not uncommon, are currently less enthusiastically discussed." But in discussion, authors compared their results with different studies. It will be good to have a table summarizes these VBT studies and lists major findings. It will be much easier for readers to have a big picture on overall VBT studies. 

Therefore, in discussion, authors will be easier to compare their findings with previous works.

Other than this suggestion, other parts are well-written. 

Author Response

Overall the paper is well-written. Considering VBT is quite a new method to treat Scoliosis, especially in some countries. More publications related to VBT will benefit the surgeons to avoid complications.

The authors explained in introduction "Non-mechanical peri-operative complications, despite not uncommon, are currently less enthusiastically discussed." But in discussion, authors compared their results with different studies. It will be good to have a table summarizes these VBT studies and lists major findings. It will be much easier for readers to have a big picture on overall VBT studies. 

Therefore, in discussion, authors will be easier to compare their findings with previous works.

Other than this suggestion, other parts are well-written. 

Author’s answer:

Thank you for your comments and for the time invested in reviewing our manuscript. While we tried to tie our results to the available literature as much as we could, the lack of similar studies makes it difficult to perform a direct comparison. For this reason, we respectfully think that adding a table with overall VBT findings would draw the attention of the reader away from the topic at hand.
However, we agree that an overview of the available data on pulmonary complications would be useful to the reader. Thus, we added a paragraph with the VBT studies that report data on pulmonary complications with the respective incidence rate. We hope that the reviewer will find this addition satisfactoty, the text now reads as follows:

“Alanay et al observed four pulmonary complications in 31 analyzed patients, which calculates an incidence of 12% 9. All patients in their series had a thoracic curve but six patients received VBT for a long thoracic curve, which usually requires splitting or dissection of the diaphragm. None of the patients had a bilateral VBT. One patient in their series was found to have a chylothorax, which required non-invasive dietary precautions, and one patient with a delayed effusion required readmission to the hospital and non-invasive medical treatment. In other similar-sized cohorts, Hoernschmeyer et al. reported one case of pneumothorax among 29 treated patients (3%) 10, while Samdani et al. observed one case of persistent atelectasis in 32 patients (3%)23. In smaller-sized groups, Newton observed one atelectasis case in 23 operated patients (4%)24 and Pehlivanoglu reported one case of chylothorax in 21 treated subjects (5%) 25. Wong et al observed five pulmonary complications in 3/5 patients (60%)26. Oppositely, Boudissa and colleagues did not observe any pulmonary complication in six operated patients. While many of the studies available in the literature showed a lower rate of pulmonary complications than the one presented in this study, other authors mainly performed thoracic VBT and the patient cohorts are thus not directly comparable.”

Round 2

Reviewer 1 Report

Part in lines 149-158 should be moved to the "material and method" part

In line 86 you write about the inclusion criteria involving the pre-op pulmonary function data - in table 2 there is a lack of this data

In lines 208-210 you mentioned about the patients coming from abroad and the ability to collect the full follow-up data. Is it possible, that they were diagnosed/eventually treated due to pulmonary complications in their country? The conclusion of the percentage of pulmonary complications seems to be underestimated.

What is the mean follow-up for the study group? also, you mentioned in lines 126-127 "The development of pulmonary perioperative complications was monitored for 3 months postoperatively; in line 208 you mentioned that your patient did not suffer from long-term complications - there is no confirmation of this conclusion in your results. Consider the change of the paper title to " early pulmonary complication..."

In risk-factor analysis - there is a lack of any statistical analysis in table 3 and 4, the data are inconclusive

Risser test also should be considered as an risk factor of pulmonary complications occurrence (p=0,007) (table 2). What about the presence of the rigid, high thoracic curve you described as a type 5 in table 3 (40%)? But there is no statistical analysis.

lines 234-236 - bypassing the diaphragm is considered as a risk factor. There is no supportive data in table 2. 

Author Response

Part in lines 149-158 should be moved to the "material and method" part

Authors’ response:

Thank you for your comment. We agree with your suggestion and the mentioned lines have been moved to the materials and methods.

In line 86 you write about the inclusion criteria involving the pre-op pulmonary function data - in table 2 there is a lack of this data

Authors’ response:

Thank you for your comment. Pre-OP PF data are shown at the bottom of Table 2.

In lines 208-210 you mentioned about the patients coming from abroad and the ability to collect the full follow-up data. Is it possible, that they were diagnosed/eventually treated due to pulmonary complications in their country? The conclusion of the percentage of pulmonary complications seems to be underestimated.

Authors’ response:

Thank you for your questions. We understand your concern, however we have a complete follow-up on all presented patients and, irrespectively of whether they do follow-ups in house or at hospitals close to their residence, all patients have been explicitly asked about pulmonary complications. Thus, we do not believe that there may be any missing data in this respect. Our complication rate is higher than that shown in other studies on thoracic curves only, but in line with the data of other groups who perform thoracolumbar and bilateral surgery. Thus, we do not believe that an underestimation of the complication rate is occurring.

What is the mean follow-up for the study group? also, you mentioned in lines 126-127 "The development of pulmonary perioperative complications was monitored for 3 months postoperatively; in line 208 you mentioned that your patient did not suffer from long-term complications - there is no confirmation of this conclusion in your results. Consider the change of the paper title to " early pulmonary complication..."

Authors’ response:

Thank you for your question. The follow-up rate is reported in the results (18.6 months), thus we respectfully chose not to change the title to “early pulmonary complications”. We agree that the wording may have been misleading, so that the paragraph was rephrased as follows:

“While pulmonary complications usually occur within the first three months after surgery, patients were monitored for this complication throughout the follow up (e. g. pulmonary function tests, specific questions at follow-up visits).”

In risk-factor analysis - there is a lack of any statistical analysis in table 3 and 4, the data are inconclusive

Authors’ response:

Thank you for your comment. We agree with you that no statistics are presented, but numbers are too small to perform any kind of analysis. For this reason, we do not present these finding as significant. However, calculating on the base of our data, a cohort of roughly 600 VBT patients would be required to have enough pulmonary complications to perform any statistics on the factors presented in Table 3 and 4. As it is highly unlikely that any VBT study group would be able to provide such a large cohort with follow-up within the next 5 years, we believe that the findings, albeit not supported by a statistical analysis, would be still of interest for the reader.

Risser test also should be considered as an risk factor of pulmonary complications occurrence (p=0,007) (table 2). What about the presence of the rigid, high thoracic curve you described as a type 5 in table 3 (40%)? But there is no statistical analysis.

Authors’ response:

Thank you for raising this point. We agree that the math would suggest the Risser grade to be a risk factor, but as age and Sanders do not show congruent results, we believe this to be likely a spurious association. This point was added to the discussion.

“While the Risser grade was higher among patients who presented a pulmonary complication, this finding is not coherent with data on age and Sanders stage. Thus, we believe that this is likely a spurious correlation”

Regarding Type 5 curves, no statistical analysis can be done as only 5 patients belonged to this cohort and numbers are to small to draw any kind of conclusion. Furthermore, type 5 patients are considered to be contraindicated for VBT (the 5 cases presented belong to the beginning of our experience, prior to the development of the classification). Thus, a further investigation on this cohort would be of little clinical significance.

lines 234-236 - bypassing the diaphragm is considered as a risk factor. There is no supportive data in table 2. 

Authors’ response:

Thank you for your comment. This analysis is presented in Table 3 and discussed in lines 205-212.
